# When Histological Tumor Type Diagnosed on Core Biopsy Changes Its Face after Surgery: Report of a Deceptive Case of Breast Carcinoma

Antonio d'Amati [1], Marta Mariano [1], Francesca Addante [1], Giovanna Giliberti [1], Giovanni Tomasicchio [2] and Mauro Giuseppe Mastropasqua [1,*]

[1]   Department of Emergency and Organ Transplantation, Section of Pathology, School of Medicine, University of Bari "Aldo Moro" 1, 70124 Bari, Italy
[2]   Department of Emergency and Organ Transplantation, Section of Breast Surgery, School of Medicine, University of Bari "Aldo Moro" 2, 70124 Bari, Italy
*   Correspondence: mauro.mastropasqua@uniba.it; Tel.: +39-0805594414

**Abstract:** The presence of stromal osteoclast-like giant cells is a distinctive feature of some rare breast tumors, accounting for less than 1% of all breast cancer cases. Although the presence of stromal osteoclast-like giant cells may be encountered in different breast tumors, some authors still describe them as a specific tumor type. Usually, a histological diagnosis of breast carcinoma is made by a pathologist through a biopsy, which is then confirmed through a surgical specimen: it is rare for the two to differ, particularly when there are pathognomonic morphological markers, such as osteoclast-like giant cells. Herein, we report a case of a 45-year-old pre-menopausal woman, who was found to have a single solid mass in her right breast on screening mammogram. She underwent a core biopsy, which showed a malignant epithelial lesion arranged in tubules, glands, and papillae, intermingled with numerous stromal osteoclast-like giant cells. Therefore, a diagnosis of breast cancer with osteoclast-like giant cells was rendered. Curiously, these cells were no longer detectable in the surgical specimen.

**Keywords:** osteoclast-like giant cells; breast carcinoma; core biopsy; surgery; histopathology





## 1. Introduction

Breast cancer with osteoclast-like stromal giant cells is considered a rare morphological variant of breast carcinoma NST (no special type) according to the WHO classification [1], whereas other authors consider it as a distinct special type [2]. In the literature, only few cases have been described, mostly in case reports [3–7].

Breast carcinomas with osteoclast-like giant cells do not show clinically relevant characteristics or peculiar features at imaging, except for peripheral hyper-vascularity at ultrasonography. However, their morphological characteristics, both macroscopic and microscopic, are unusual. In fact, the macroscopic aspect is so unique that it can be recognized and suspected on visual inspection. It shows brown or red brown color and bulges above the surrounding parenchyma when cut on fresh tissue. In formalin-fixed specimens, the color tends to be darker, also raising suspicion for metastatic melanoma.

Histologically, the main feature is the presence of many giant cells in the stroma, or even in the glandular lumina, which, however, are not considered neoplastic [1–7]. The giant cells are large, with abundant cytoplasm and containing numerous centrally located nuclei, sometimes with evident nucleoli. The carcinoma cells may be arranged in tubules, glands, and papillae, and may also show mucinous or lobular patterns. Several inflammatory cells, such as lymphocytes, monocytes, and histiocytes, are detectable in the stroma, along with many extra-vascular red blood cells and hemosiderin deposits, which are definitively responsible for the peculiar brownish color of the tumor.

## 2. Case Presentation

A 45-year-old lady, during her screening ultrasonography, was found to have an 8-millimeter solid mass in the upper outer quadrant of right breast, characterized by irregular margins and inhomogeneous internal echoes, suspicious for carcinoma. An ultrasound-guided core needle biopsy of the lesion was performed and a diagnosis of gland-forming breast carcinoma with many erythrocytes, hemosiderin, and stromal osteoclast-like giant cells was rendered. The neoplastic cells showed a moderate grade of atypia. No further assays were performed (Figures 1 and 2).

Two weeks later, the patient underwent a quadrantectomy with sentinel lymph-node removal, as recommended by the multidisciplinary tumor board.

The sentinel lymph node was examined intraoperatively with the One-Step Nucleic Acid Amplification technique, and the result was negative.

The breast specimen was also sent for the intraoperative evaluation of its margins and, on the cut surface, the lesion showed a red–brown color, without showing any bulging. Furthermore, 24 h later, after fixation, the lesion did not show the expected peculiar macroscopic characteristics and color. The entire lesion and the surrounding normal breast tissue were extensively sampled for histological examination.

Microscopically, the lesion showed a fibrous reaction at the site of the previous core biopsy, along with adjacent foci of ductal carcinoma in situ and a small (3-millimeter) residual part of invasive carcinoma, grade 2. Surprisingly, even after a meticulous examination, osteoclast-like giant cells were no longer detected in the surgical specimen (Figures 3 and 4).

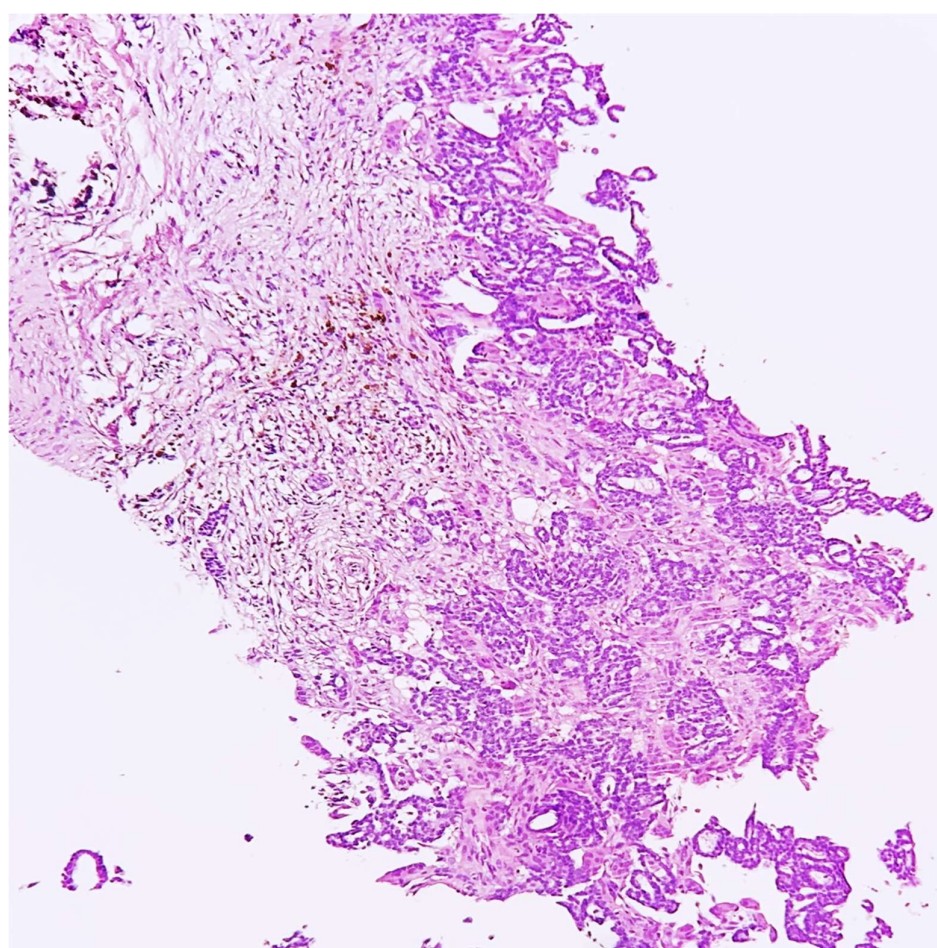

**Figure 1.** Ultrasound-guided core needle biopsy showing invasive breast carcinoma, associated with hemosiderin deposition (HE, 100×).

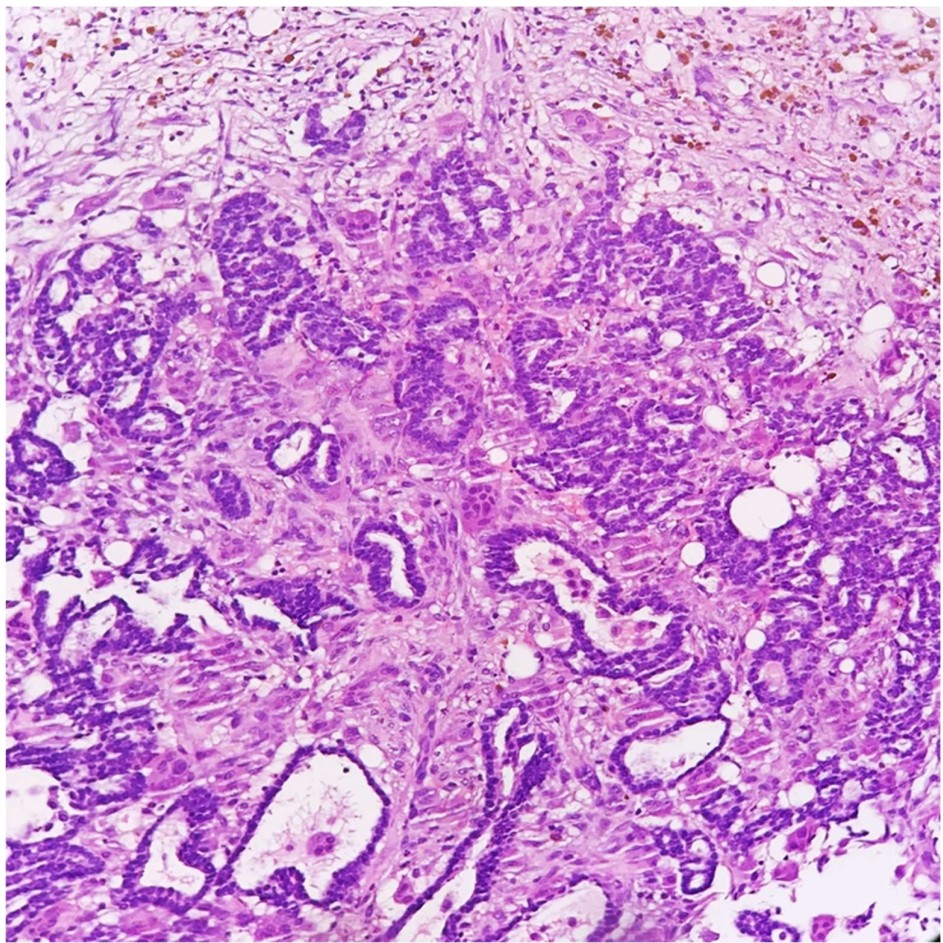

**Figure 2.** Ultrasound-guided core needle biopsy showing invasive gland-forming carcinoma, associated with erythrocytes, hemosiderin, and many stromal osteoclast-like giant cells (HE, 200×).

Immunohistochemical analyses for prognostic and predictive markers were performed on the surgical specimen. The tumor showed high expression of estrogen receptors (>95% tumor cells), but was negative for progesterone receptors with a positive internal control. Ki67 labeling index was 8% and there was no HER2 over-expression (score of 0, according to ASCO/CAP 2018 guidelines) [8,9]. The CD68 immunostaining was almost negative, excluding some CD68-positive stromal macrophages, confirming the absence of stromal osteoclast-like giant cells (Figure 5).

Considering the results after the histological examination of the breast specimen, we decided to re-evaluate the pre-operative core biopsy. The re-evaluation of the former core biopsy confirmed the presence of many stromal osteoclast-like giant cells (Figure 6), the high positivity of tumor cells for estrogen receptors and the negativity for HER2.

Moreover, a diffuse immunoreactivity for progesterone receptors (>95% tumor cells) (Figure 7), in contrast with the surgical-specimen finding, and a higher Ki67 labeled index, of 15%, were observed, as already reported in a larger published series [7].

The CD68 immunostaining intensely highlighted all the present osteoclast-like giant cells, confirming their well-known monocyte-lineage origin.

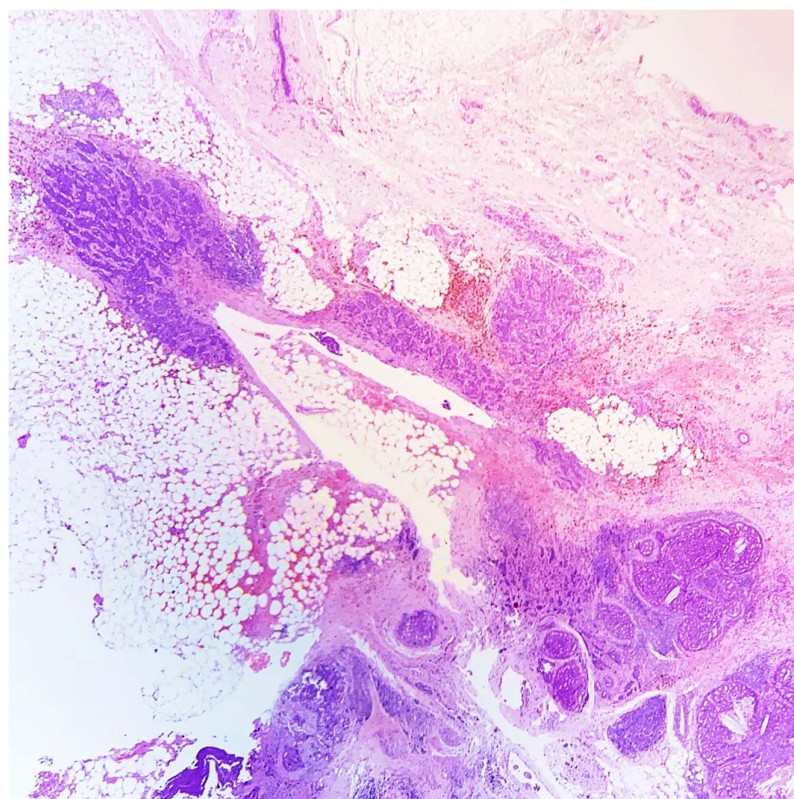

**Figure 3.** Histological section from quadrantectomy specimen showing fibrous reaction in the site of previous biopsy, along with foci of ductal carcinoma in situ and residual invasive carcinoma (HE, 25×).

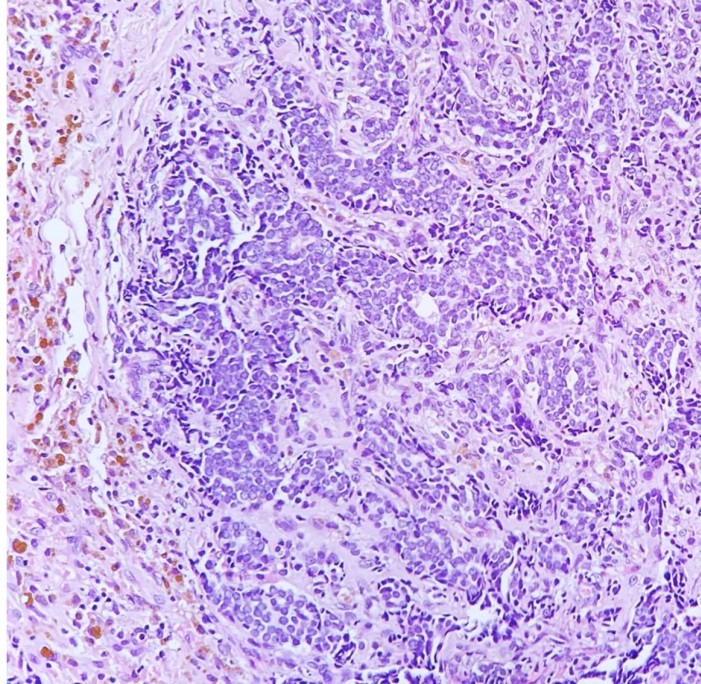

**Figure 4.** Histological detail of surgical specimen showing invasive nests of neoplastic cells and hemosiderin deposition; stromal osteoclast-like giant cells were no longer detectable (HE, 200×).

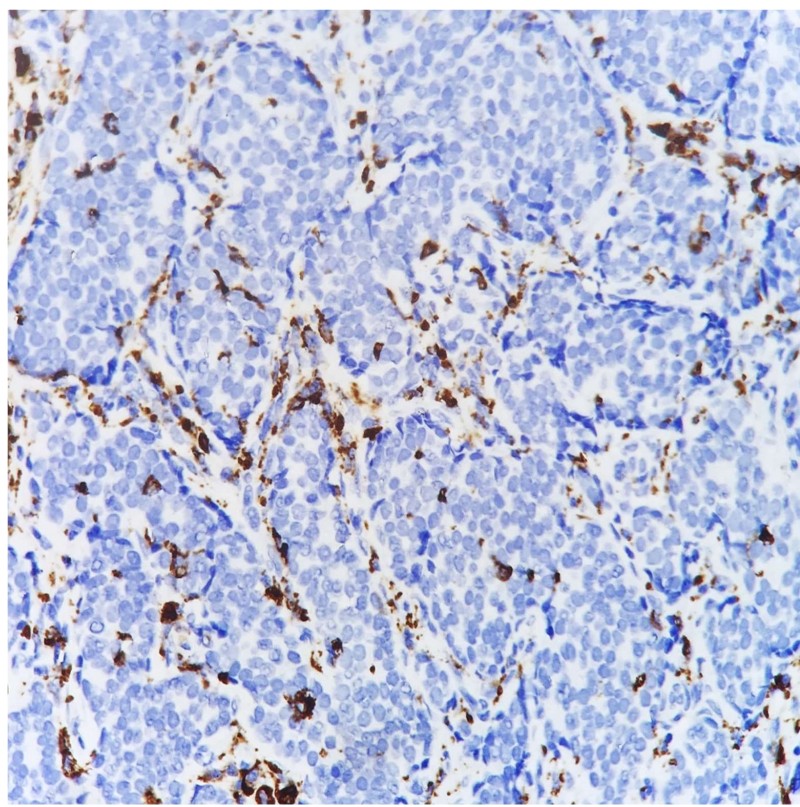

**Figure 5.** Histological detail of surgical specimen showing absence of stromal osteoclast-like giant cells, demonstrated by immunoreaction for CD68. Many CD68-positive macrophages were still detectable (400×).

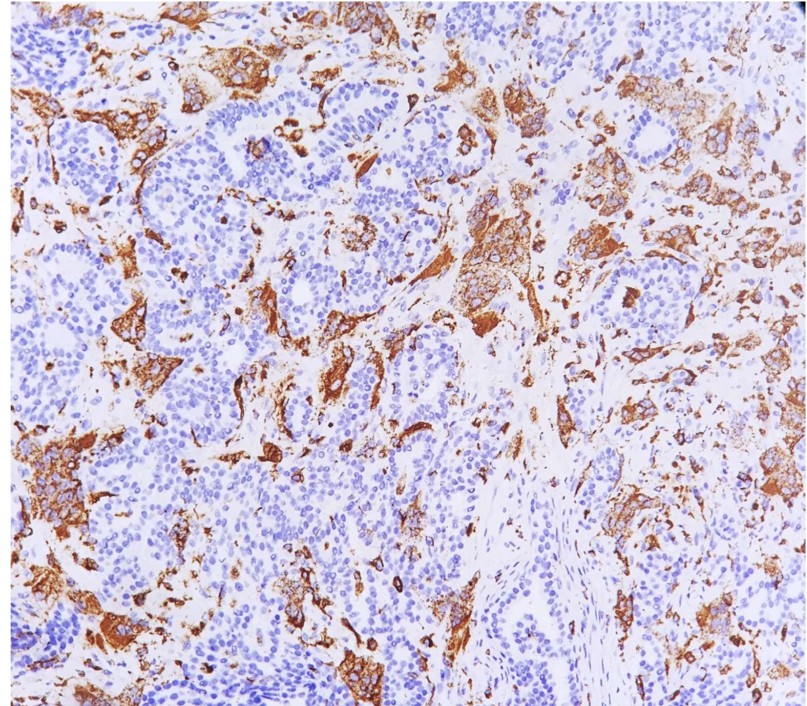

**Figure 6.** Immunohistochemical reaction for CD68 in presurgical biopsy showing positive macrophages and numerous stromal osteoclast-like giant cells, intermingled with CD68-negative neoplastic cells (200×).

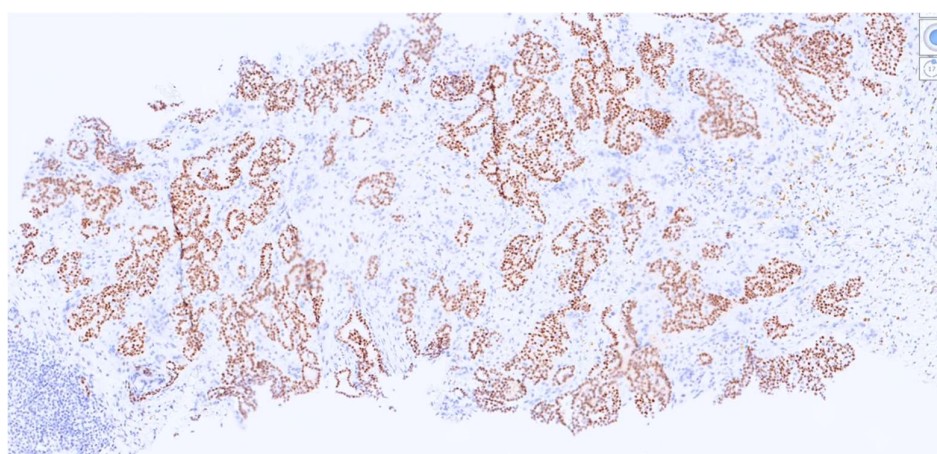

**Figure 7.** Immunoreactivity for progesterone receptor in almost all neoplastic cells of the presurgical biopsy (100×).

## 3. Discussion

The presence of stromal osteoclastic-like giant cells intermingled between breast cancer cells is an infrequent histological finding, and its uncertain clinical implications are not fully understood, partly due to the current rarity of relevant studies in the literature [7].

Macroscopically, it is commonly described as a round, well-circumscribed nodule, showing a pathognomonic dark brown color [4]. However, in our case, the peculiar macroscopic aspect appeared to be lost, probably because of the previous diagnostic procedures and the small size of residual neoplasm.

Indeed, the peculiarity of the case reported herein is in the different histological appearances of the core biopsy and the surgical specimen, characterized by the absence of stromal osteoclast-like giant cells, which were present instead in the pre-operative biopsy. This is the first case in which the core biopsy provoked the removal of a peculiar feature of the tumor, such as the presence of osteoclast-like giant cells, thus leading to a different histological diagnosis after surgery. Even though it is not possible to ascertain the exact mechanisms that lead to giant cells disappearance, many hypotheses could be considered.

The absence of giant cells may be relatable to intratumoral heterogeneity, and the possibility that these cells could have been present in only a part of the tumor should not be excluded; nevertheless, they are more frequently scattered throughout the tumor. Moreover, in the surgical specimen, there was only a minimal amount of invasive cancer, and most of the invasive neoplasm was present in the biopsy specimen. Therefore, in our opinion, it is possible that the few giant cells present in the tumor were removed after the biopsy. Considering the strict adherence to appropriate processing procedures, we excluded fixation issues or other technical issues as possible explanations. The same was concluded with regards to eventual stromal changes, since a short period of time elapsed between the biopsy and the surgical resection.

However, even though this is the first reported case featuring the disappearance of osteoclast-like giant cells after biopsy, many cases in the literature which report differences in histologic findings between surgical and pre-operative specimens, particularly regarding immunohistochemistry [10,11].

In fact, our case also showed a loss of expression for the progesterone receptors in the surgical specimen, inducing oncologists to consider it as a less hormone-responsive tumor and to suggest the presence of a more aggressive behavior [12].

The pathogenic and prognostic significance of osteoclastic-like giant cells is still debated, as there are controversial reports [5], likely due to their marked heterogeneity [7].

Recently, it has been described that osteoclastic-like giant cells share phenotypic relationships with monocytes, as demonstrated by CD68 expression, but there is no comparison in terms of miRNA expression pattern, which, on the contrary, shows a greater similarity

with neoplastic cells [13]. This could suggest that osteoclastic-like giant cells may have important pathogenic and prognostic roles, possibly influencing patients' responses to the treatment options that are currently recommended.

## 4. Conclusions

Histological evaluation after core biopsy represents the gold standard for pre-operative breast cancer diagnosis, which is usually confirmed after surgical-specimen examination. However, some features may occasionally vary between the core biopsy and the surgical specimen. Very rarely, as described in our case, core biopsies may also cause the disappearance of peculiar histological characteristics, leading to a different diagnosis upon the examination of the surgical specimens. Therefore, it is always fundamental for pathologists to keep these aspects in mind, carefully observing surgical specimens and eventually reevaluating core biopsy slides when a different finding appears, in order to obtain the correct diagnosis and to ensure the best treatments for patients.

**Author Contributions:** Conceptualization, M.M. and M.G.M.; methodology, G.G.; software, G.T.; validation, A.d., M.M. and F.A.; formal analysis, G.T.; investigation, G.G.; resources, G.G.; data curation, G.T.; writing—original draft preparation, A.d., M.M. and F.A.; writing—review and editing, A.d. and M.G.M.; visualization, F.A.; supervision, M.M. and M.G.M. All authors have read and agreed to the published version of the manuscript.

**Funding:** This research received no external funding.

**Institutional Review Board Statement:** This study did not require ethical approval.

**Informed Consent Statement:** Written informed consent was obtained from the patient to publish this paper.

**Conflicts of Interest:** The authors declare no conflict of interest.

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
