# Peer review of "When Histological Tumor Type Diagnosed on Core Biopsy Changes Its Face after Surgery: Report of a Deceptive Case of Breast Carcinoma"

_reports, doi:10.3390/reports5040038_

Round 1

Reviewer 1 Report

In this manuscript, the authors presented a particular case of a 45-year-old breast cancer patient, from whom the lesion was removed completely before the surgery and no distinctive pathological features were detected from the surgery specimen samples. Although there were HE staining, histological and immunohistochemistry results to support the conclusion, there are still some concerns need to be addressed before considering publication, listed below:

Major Concerns:

1.                The abstract was not well written. Some background introduction was unnecessary in the abstract, and the case discussed in the manuscript was not well summarized and concluded.

2.                The surgery was two weeks after ultrasound-guided core needle biopsy, is there any possibility that the disease status might have changed after the intervention?

3.                Have you ruled out possibilities of technical mishandling when prepared the specimen samples from surgery?

4.                On page 3 line 65 “the lesion showed a red brown color consistent with the previous core biopsy”, this was the only consistency from core biopsy and surgery specimen, have you ruled out possibilities of labeling errors? More diagnosis is necessary to confirm the disease status during the surgery. If only one lesion was observed and completely removed during biopsy, was there any other checks that were done before the surgery? Was there a reason for the further surgery?

Minor Comments:

1.                On page 1 line 16, “Authors” should be lowercase.

2.                Please list the abbreviation for “NST” on page 1 line 30.

3.                Please check grammar on page 1 line 33-34 “Clinically, these tumors are like other breast carcinomas, although their morphological characteristics, both macroscopic and microscopic, are unusual”.

4.                Please check grammar on page 8 line 130-131 “Core biopsies may not only completely remove the lesion, but as in this case, also the distinctive features”.

Author Response

Dear Reviewer,

we would really like to thank you for the opportunity to resubmit our article and for your helpful comments, as we considered them as precious advices that allowed us to improve the quality of our manuscript.

1. The abstract was not well written. Some background introduction was unnecessary in the abstract, and the case discussed in the manuscript was not well summarized and concluded.

We appreciate reviewer’s suggestions. We modified the abstract and improved manuscript according to the suggestions.

2. The surgery was two weeks after ultrasound-guided core needle biopsy, is there any possibility that the disease status might have changed after the intervention?

We would like to thank the reviewer for the interesting question. According to the literature and considering the shortness of the period between biopsy and surgical resection, there are very low probabilities that the disease status might have changed after the intervention.

3. Have you ruled out possibilities of technical mishandling when prepared the specimen samples from surgery?

Every step during the sampling has been done and verified by two expert pathologists. The surgical specimen was carefully observed and described during intraoperative examination and then completely sampled, in order to examine any part of cancer and normal tissue histologically

4. On page 3 line 65 “the lesion showed a red brown color consistent with the previous core biopsy”, this was the only consistency from core biopsy and surgery specimen, have you ruled out possibilities of labeling errors? More diagnosis is necessary to confirm the disease status during the surgery. If only one lesion was observed and completely removed during biopsy, was there any other checks that were done before the surgery? Was there a reason for the further surgery?

We thank the reviewer for the opportunity to clarify this important point. Every labeling error was excluded with certainty, we checked and reviewed every step in the specimen processing procedure. The core needle biopsy was made in order to obtain a histological diagnosis after radiological examination, which identified an opacity highly suspicious for cancer. After the biopsy, histologically and radiologically is not possible to predict the eventual and infrequent full removal of the lesion. In our case, the biopsy did not allow complete removal of the lesion, because in the breast specimen part of invasive cancer was still detectable (as demonstrated in Fig 3). According to international protocols and guidelines, after biopsy and histological diagnosis, the patient underwent surgical resection without any further diagnostic assessments and only the surgical resection allowed the complete removal of the cancer. The peculiarity of our case is represented by the disappearance of osteoclast-like giant cells in breast specimen, two weeks after biopsy, suggesting a change of histotype or the fact that the presence of these cells may not be indicative of a specific histologic subtype.

Minor Comments:

  1. On page 1 line 16, “Authors” should be lowercase.
  2. Please list the abbreviation for “NST” on page 1 line 30.
  3. Please check grammar on page 1 line 33-34 “Clinically, these tumors are like other breast carcinomas, although their morphological characteristics, both macroscopic and microscopic, are unusual”.
  4. Please check grammar on page 8 line 130-131 “Core biopsies may not only completely remove the lesion, but as in this case, also the distinctive features”.

Amended

Reviewer 2 Report

This is a single case report on breast carcinoma with osteoclast-like giant cells pointing out major differences in the histological features of the same tumor in core biopsy versus in surgical specimen. Unfortunately, a thorough discussion on the possible reasons for this discrepancy is missing, although this is the only unusual moment in this report and could be very interesting for the readers and for the everyday practice.

1. A thorough discussion about the reasons of the discrepancies between the tumor features in core biopsy specimen and in the surgical specimen would improve the manuscript considerably and shift the focus of the manuscript from a simple case demonstration to a general issue in histopathology. Is this a result of intratumoral heterogeneity (differences in progesterone receptor expression may favor this possibility, although progesterone positive tumor cells were present at the margin of the core biopsy indicating rather a technical issue regarding the surgical specimen)? Or the tumor stroma has undergone changes during the two weeks between the core and the surgery? Could the discrepancies relate to fixation or other technical issues? Complete removal of all the osteoclast-like giant cells with the core biopsy, as the authors suggest, is also a possibility, however, these giant cells are usually dispersed throughout the tumor. (This reviewer has a similar experience with a case in which the giant cells were only few in the surgical specimen, but present and were found after a long search.) Please try to find supporting references to this important subject (the literature on core biopsy - surgical specimen discrepancies is rather abundant, focus more on discrepancies in histological tumor type) .

2. Importantly, every word in the text has to be evaluated again. Most of the text is grammatically correct, but it does not really reflects the aim of the writer. I will illustrate this with some examples, but the same is evident in the entire text. Please read and revise carefully the entire text this way.

- Lines 12 -13. "could be so abundant to represent a distinct histotype..." - first of all, the number of the stromal giant cells is not a criterion in these cases; "could be" means that it could but it is not; the giant cells themselves are not the tumor, thus they not "represent" rather indicate a distinct tumor type. 

- line 13:  "a rare frequency accounting about 1%" - 1% is rare itself, the frequency is rather low or high, not rare

- Lines 12-13: I would say: Presence of stromal osteoclast-like giant cells is a distinctive feature of some rare beast tumors that account for less than 1% of all breast cancer cases... (or a similar formulation). 

3. I suggest the authors to change the title: "difficult to demonstrate" means that at the end you succeeded to demonstrate this although it was difficult; "at surgery " means "during the surgical intervention" - it should be "surgical specimen". The diagnosis is usually "breast carcinoma" and histological tumor typing is part of diagnosis or of the description. So, if you wish to keep the style of the title, my proposal would be: When histological tumor type in preoperative core biopsy differs from that in postoperative surgical specimen: report of a case of breast carcinoma

4. In In addition, the legends to the figures should clearly indicate whether the figures illustrate the core or the surgical specimen, as the last images come after the statement of lacking giant cells in the surgical specimen followed immediately by figure 6 illustrating the presence of them.

5. The Conclusions are also problematic: the first of them has nothing to do with the demonstrated case, thus it should be skipped. The second one has to be modified in order to reflect the widened discussion as proposed above.

6. Figure 4., 9 o'clock at the tumor periphery, there is a group of small "empty" nuclei surrounded with some cytoplasm - could it be a giant cell ?

Author Response

Dear Reviewer,

we would really like to thank you for the opportunity to resubmit our article and for your helpful comments, as we considered them as precious advices that allowed us to improve the quality of our manuscript.

1. A thorough discussion about the reasons of the discrepancies between the tumor features in core biopsy specimen and in the surgical specimen would improve the manuscript considerably and shift the focus of the manuscript from a simple case demonstration to a general issue in histopathology. Is this a result of intratumoral heterogeneity (differences in progesterone receptor expression may favor this possibility, although progesterone positive tumor cells were present at the margin of the core biopsy indicating rather a technical issue regarding the surgical specimen)? Or the tumor stroma has undergone changes during the two weeks between the core and the surgery? Could the discrepancies relate to fixation or other technical issues? Complete removal of all the osteoclast-like giant cells with the core biopsy, as the authors suggest, is also a possibility, however, these giant cells are usually dispersed throughout the tumor. (This reviewer has a similar experience with a case in which the giant cells were only few in the surgical specimen, but present and were found after a long search.) Please try to find supporting references to this important subject (the literature on core biopsy - surgical specimen discrepancies is rather abundant, focus more on discrepancies in histological tumor type).

1. We would really like to thank the Reviewer for the precious suggestions and advices, that allowed us to improve the discussion section of the manuscript. Effectively, the absence of giant cells may also be relatable to intratumoral heterogeneity and is not possible to exclude that these cells could have been present just in a part of the tumor. Moreover, in the surgical specimen there was only a minimal part of invasive cancer and most of the invasive neoplasm was present in the biopsy specimen. So, in our opinion, it is possible that the few giant cells present in the tumor were removed after biopsy. We exclude fixation and other technical issue, as the same as we think that significant stromal changes could be excluded because of the short period between biopsy and surgery. As suggested by the reviewer, there is abundant literature regarding discrepancies between biopsy and specimen and we included the most relevant as references in discussion section.

2. Importantly, every word in the text has to be evaluated again. Most of the text is grammatically correct, but it does not really reflects the aim of the writer. I will illustrate this with some examples, but the same is evident in the entire text. Please read and revise carefully the entire text this way.

- Lines 12 -13. "could be so abundant to represent a distinct histotype..." - first of all, the number of the stromal giant cells is not a criterion in these cases; "could be" means that it could but it is not; the giant cells themselves are not the tumor, thus they not "represent" rather indicate a distinct tumor type. 

- line 13:  "a rare frequency accounting about 1%" - 1% is rare itself, the frequency is rather low or high, not rare

Lines 12-13: I would say: Presence of stromal osteoclast-like giant cells is a distinctive feature of some rare breast tumors that account for less than 1% of all breast cancer cases... (or a similar formulation).

2. We thank the reviewer. We amended the suggested lines and carefully revised the entire text of the manuscript.

3. I suggest the authors to change the title: "difficult to demonstrate" means that at the end you succeeded to demonstrate this although it was difficult; "at surgery " means "during the surgical intervention" - it should be "surgical specimen". The diagnosis is usually "breast carcinoma" and histological tumor typing is part of diagnosis or of the description. So, if you wish to keep the style of the title, my proposal would be: When histological tumor type in preoperative core biopsy differs from that in postoperative surgical specimen: report of a case of breast carcinoma.

3. Thank you for the suggestion. We modified the title of the manuscript accordingly.

4. In addition, the legends to the figures should clearly indicate whether the figures illustrate the core or the surgical specimen, as the last images come after the statement of lacking giant cells in the surgical specimen followed immediately by figure 6 illustrating the presence of them.

4. Thank you for the suggestion. We modified the legends of the figures, clearly indicating whether images belong to biopsy or surgical specimen.

5. The Conclusions are also problematic: the first of them has nothing to do with the demonstrated case, thus it should be skipped. The second one has to be modified in order to reflect the widened discussion as proposed above.

5. We thank the reviewer. We improved conclusions according to the suggestions.

6. Figure 4., 9 o'clock at the tumor periphery, there is a group of small "empty" nuclei surrounded with some cytoplasm - could it be a giant cell?

6. We thank the reviewer for the question. As suggested by the reviewer, we also interpreted that group of nuclei as a possible giant cell, but immunohistochemistry showed negativity for CD68.

Round 2

Reviewer 1 Report

Dear Authors, 

The revised manuscript was significantly improved. 

Thanks,

Author Response

Thank you for your comments.

Reviewer 2 Report

The revision improved the quality of the manuscript substantially. I just suggest to make minor modifications in the legends to the figures. If the legend starts with "Surgical specimen" then I expect a macrophoto of it. It should start with "Histological detail of the surgical specimen slide" or something similar.  Otherwise, the manuscript is acceptable for publication, no further review is needed.

Author Response

Thank you for your comments. We have revised accordingly.